# LaneCMKT: Boosting Monocular 3D Lane Detection with Cross-Modal Knowledge Transfer

Runkai Zhao
rzha9419@uni.sydney.edu.au
The University of Sydney
Sydney, Australia

Heng Wang
hwan9147@uni.sydney.edu.au
The University of Sydney
Sydney, Australia

Weidong Cai
tom.cai@sydney.edu.au
The University of Sydney
Sydney, Australia

## ABSTRACT

Detecting 3D lane lines from monocular images is garnering increasing attention in the Autonomous Driving (AD) area due to its cost-effective edge. However, current monocular image models capture road scenes lacking 3D spatial awareness, which is error-prone to adverse circumstance changes. In this work, we design a novel cross-modal knowledge transfer scheme, namely **LaneCMKT**, to address this challenge by transferring 3D geometric cues learned from a pre-trained LiDAR model to the image model. Performing on the unified Bird's-Eye-View (BEV) grid, our monocular image model acts as a student network and benefits from the spatial guidance of the 3D LiDAR teacher model over the intermediate feature space. Since LiDAR points and image pixels are intrinsically two different modalities, to facilitate such heterogeneous feature transfer learning at matching levels, we propose a dual-path knowledge transfer mechanism. We divide the feature space into shallow and deep paths where the image student model is prompted to focus on lane-favored geometric cues from the LiDAR teacher model. We conduct extensive experiments and thorough analysis on the large-scale public benchmark *OpenLane*. Our model achieves notable improvements over the image baseline by **5.3%** and the current BEV-driven SoTA method by **2.7%** in the F1 score, without introducing any extra computational overhead. We also observe that the 3D abilities grabbed from the teacher model are critical for dealing with complex spatial lane properties from a 2D perspective.

## CCS CONCEPTS

• **Computing methodologies** → **Transfer learning**; **Computer vision**.

## KEYWORDS

Multi-modality, Transfer Learning, 3D Vision, Lane Detection

**ACM Reference Format:**
Runkai Zhao, Heng Wang, and Weidong Cai. 2024. LaneCMKT: Boosting Monocular 3D Lane Detection with Cross-Modal Knowledge Transfer. In *Proceedings of the 32nd ACM International Conference on Multimedia (MM '24), October 28-November 1, 2024, Melbourne, VIC, AustraliaProceedings of the 32nd ACM International Conference on Multimedia (MM'24), October 28-November 1, 2024, Melbourne, Australia.* ACM, New York, NY, USA, 10 pages. https://doi.org/10.1145/3664647.3681038

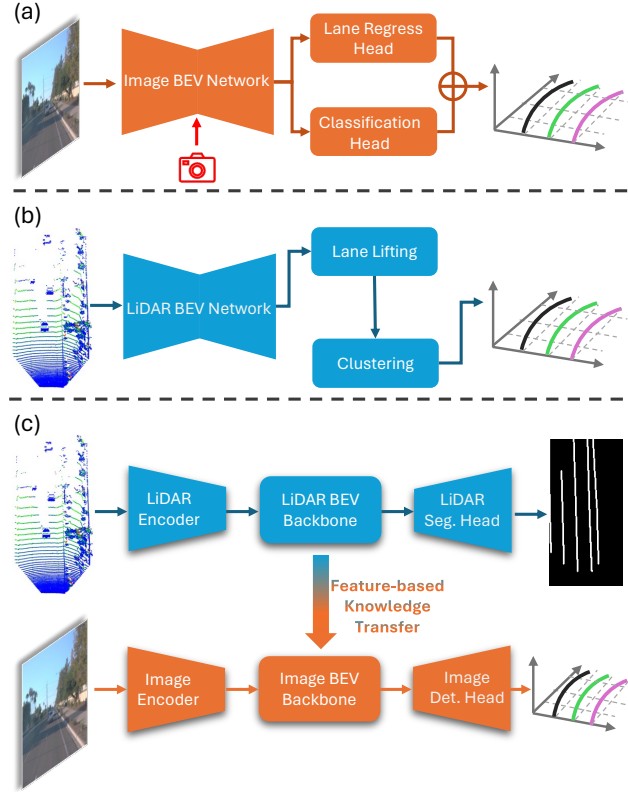

**Figure 1:** *3D lane detection frameworks with different modalities.* **Both image- (a) and LiDAR- (b) based 3D lane detection models utilize the BEV representation learning to globally delineate lane layouts and encode modality-specific features for the following individual lane inference. Based on this, we utilize this unified perspective space to design a cross-modal knowledge transfer scheme, imbuing image BEV features with spatial awareness derived from LiDAR features.**

## 1 INTRODUCTION

3D lane detection, aiming at localizing lane lines within the 3D ego-vehicle coordinate system, plays a vital role in the driving scene understanding. This technique is integral to various applications related to vehicle control [36] and high-definition map reconstruction [20, 21]. Although LiDAR-based methods achieve remarkable detection accuracy, there is a burgeoning interest in vision-centric methods from both academia and industry. Compared to the complex sensor suite required for LiDAR, cameras are significantly more

cost-effective and are financially favored for mass production in the Autonomous Driving (AD) industry. Moreover, camera images provide finer visual cues through dense pixel representations, which are especially critical for detecting slender and long-spanning lanes.

To bridge 2D and 3D space in monocular 3D lane detection, most studies establish a dense grid correspondence to transpose front-viewed image features into the intermediate Bird's Eye View (BEV) space via Inverse Perspective Mapping (IPM) [5, 8, 9, 12, 18, 37, 43]. Aligning the image BEV feature with a lane anchor map allows for the projection of detected lanes back into 3D space using geometric regression and height estimation, as illustrated in Figure 1 (a). While IPM-based methods offer an end-to-end framework for 3D lane detection, they cannot enable the monocular model to acquire accurate spatial information, which limits their 3D lane detection performance. The reasons are analyzed as follows: (1) IPM inherently cannot provide depth information, which is crucial for a monocular model to accurately estimate distances and elevations within 3D space; (2) IPM presupposes a flat ground assumption that does not hold consistent for real-world driving scenarios (e.g. uphill/downhill), leading to distortions onto the image BEV feature and deviations from the actual 3D geometry; (3) During the IPM-based model training, spatial awareness learning is solely guided by basic lane anchor regression, which is insufficient to direct the monocular model to learn representative features being sensitive to the 3D world.

On the other hand, the LiDAR point cloud offers explicit spatial information to complement the monocular model for 3D lane detection. Recently, some works have started to explore the combination of image pixel and LiDAR point features for performance improvement. $M^2$-3DLane [25] utilizes depth estimation to project image frontal feature into 3D space, subsequently aggregating pseudo-lidar image and lidar features in the BEV space. Unlike conventional query-based methods in object detection [4, 47], enriching learnable query embedding with more task-specific information [11, 22] could promote model performance and training convergence. Building on this, DV-3DLane [24] integrates modality features of image and LiDAR data at the query level and then processes this cross-modal query set into a transformer decoder for further prediction. While combining feature representations from multiple resources improves lane detection accuracy, the addition of extra modality input increases computational cost and the burden of extra capturing devices during inference. Moreover, the high requirement for calibration and synchronization across sensors restricts the applicability of the multi-modal fusion method.

Inspired by advancements seen in AD visual tasks through cross-modal distillation [6, 7, 10, 32, 39–42], we design a novel training scheme to boost monocular 3D **Lane** detection with **C**ross-**M**odal **K**nowledge **T**ransfer, namely **LaneCMKT**. Lane feature representations in the image model are guided to assimilate the non-homogeneous features extracted from a pre-trained LiDAR model, thereby incorporating 3D geometric awareness into image features while maintaining the semantic details essential for precise lane detection. To handle the modality discrepancy in multi-layer features, we design a dual-path knowledge transfer mechanism at shallow and deep levels to enhance the explicitness of cross-modal knowledge transfer. Shallow features, rich in modality-specific information, highlight the inherent modality gap between

point clouds and images in the initial model layers. Directly aligning these features of a large modality gap introduces disturbance into the image feature learning and diminishes the effectiveness of knowledge transfer. To address this issue, we design an adaptive scaling strategy with quantifying lane geometric properties, which enables the image model to selectively leverage beneficial LiDAR features for lane instances. While the model features are encoded into the modality-agnostic latent embedding, the alignment of deep abstract features allows the image student model to directly access 3D contextual scene understanding from the LiDAR teacher model. The experiments demonstrate that our monocular lane model can significantly benefit from our proposed cross-modal knowledge transfer framework, highlighting marked improvements in lane detection accuracy and geometric regression.

Our main contributions can be summarized as follows:

- We present LaneCMKT, a novel training scheme for monocular 3D lane detection that enables effective cross-modal knowledge transfer in BEV space, leveraging the spatial insights from LiDAR data to enhance image feature learning.
- We design a dual-path knowledge transfer mechanism to accurately extract multi-layer geometric knowledge from the LiDAR teacher model. To mitigate the modality discrepancy at shallow level features, we propose an adaptive scaling strategy that encourages the image student model to focus on crucial lane geometry under the supervision of LiDAR point features.
- We conduct extensive experiments and analysis on a large-scale lane dataset, *OpenLane*. Without bells and whistles, our proposed method outperforms the baseline by **5.3%** and the BEV-driven SoTA method by **2.7%** in F1 score.

## 2  RELATED WORKS

### 2.1  Cross-Modal Knowledge Transfer for Autonomous Driving

Knowledge transfer is a proven useful method for model compression while maintaining high accuracy [28, 29, 33, 44]. As deep learning techniques offer a pathway to interact cross-modal information in the latent feature space [13, 16, 19, 30, 34, 35, 46], cross-modal knowledge transfer attract increasing attention for solving multi-modal visual tasks. Typically, a well-trained, complex teacher model processes the input to generate informative representations which then guide the feature learning or output logits of a student model that receives input from a different modality. This approach can be applied to AD visual tasks.

The LiDAR model acts as the teacher, providing explicit spatial cues not readily available from image data alone. MonoDistill [7] builds an image-like LiDAR teacher model by projecting LiDAR points onto the image plane. To bridge the modality gap between LiDAR and image data, it transfers structural cues from teacher to student by maintaining similar local region affinity relations. CMKD [10] pioneers BEV-based knowledge transfer that enables the student model's features to emulate LiDAR BEV representation. The process of distilling classification logits offers a rich category distribution at the output end, facilitating knowledge sharing between models. However, distillation from different input data

modalities may introduce noise and impair performance. To address this, BEVDistill [6] selectively weights the predictions of each query in the LiDAR model and employs a quality control score to estimate instance-wise importances for maximizing their mutual information. DistillBEV [40] enhances feature transfer adaptability by incorporating spatial attention and scale awareness. Besides the uni-directional knowledge transfer from LiDAR points to image pixels, the RGB image provides semantic cues to counteract the sparsity of the LiDAR point cloud. 2DPass [42] utilizes 2D semantic priors to assist LiDAR semantic segmentation, training both student and teacher models concurrently to distill multi-modal knowledge into a single-modal feature. Recognizing that the number of pixels in an image vastly outnumbers the points in a point cloud, ProtoTransfer [32] aims to fully exploit image data by integrating unmatched 2D and original 3D and fused features into a prototype bank. This unified space facilitates the transfer of class-aware knowledge to LiDAR point features, enhancing the overall effectiveness of knowledge distillation across modalities.

## 2.2 3D Lane Detection

3D lane detection in a vision-centric manner attracts more and more research attention, but accurately reconstructing 3D information from single-view images remains challenging. The prevalent methods apply IPM to transform 2D frontal-view space to BEV which offers clearer geometric properties of lanes [5, 8, 9, 12, 18, 37, 43]. 3D-LaneNet [8] and Gen-LaneNet [9] project image features into BEV space and regress the anchor offsets of lanes with space alignment in the 3D lane detection head. Persformer [5] adopts a query-based method to adaptively construct BEV features with a deformable attention mechanism. Despite these advancements in the end-to-end lane detection framework, due to the inherent lack of monocular images, the IPM-based methodologies cause ambiguity in BEV features of lane layout and compromise its robustness [5, 18, 24]. To address this, Anchor3Dlane [12] directly samples from original FV image features to retain richer context information with iterative anchor regression. CurveFormer [1], on the other hand, converts a set of lane anchor points into learnable queries and employs a curve cross-attention decoder to parametrically represent lane geometry. These anchor-driven approaches improve the utilization of lane-favored spare image features but still struggle to bridge the gap between 2D and 3D spaces effectively.

In light of these limitations, some studies attempt to reformulate this task as a LiDAR-based BEV segmentation [14, 27, 45]. LiLaDet [45] predicts semantic lane map on the BEV plane and then applies sparse voxel techniques for geometric regression and 3D lane curve fitting. Although LiDAR-based representations provide precise localization information, their sparsity limits advancements in semantic instance understanding. To overcome this, recent research has investigated multimodal strategies that integrate synthesized lane information from both images and point clouds [24, 25]. $M^2$-3Dlane [25] leverages LiDAR point clouds to lift image features into 3D space and combines multimodal features within the BEV space. Additionally, DV-3DLane [24] employs an attention-based matching strategy to enhance image instance-level queries with LiDAR point features. Unlike previous works focusing on model design and multi-modal fusion, our method designs a novel cross-modal knowledge transfer framework to enhance image feature learning

to access 3D spatial cues of a pre-trained LiDAR teacher model in the training phase. During the inference phase, our approach endows the lightweight image model with vital spatial awareness without necessitating additional modality input or extra computational costs.

## 3 METHODOLOGY

In this section, we delve into the details of our proposed LaneCMKT framework. An overview of our framework is provided in Figure 2, where we outline the general workflow of the image student model and the pre-trained LiDAR teacher model, as discussed in Section 3.1. Following this, we demonstrate the dual-path knowledge transfer mechanism, for shallow and deep level features, to allow effective multi-layer geometric features of LiDAR data to guide image feature learning, detailed in Section 3.2. The supervision losses of our cross-modal transfer training are illustrated in Section 3.3.

### 3.1 Overall Framework

**LiDAR Model.** LiDAR BEV segmentation is approved to capture global contextual knowledge that can accurately produce a semantic map for comprehensive top-view scene understanding [45]. This finding motivates us to distill this valuable spatial knowledge to benefit monocular image features. Given a set of LiDAR points $P \in \mathcal{R}^{N \times 4}$ including the $x$, $y$, $z$ coordinates and intensity of each point, it is initially partitioned into a grid of pillars [17]. These pillar features are scattered down into the 2D LiDAR BEV features that are then forwarded into a U-shaped architecture to perform semantic map segmentation. As depicted in Figure 2, the intermediate LiDAR BEV features extracted from the backbone are leveraged as feature imitation targets to guide the image-based BEV features learning, which are denoted as $F_{bev,m}^L \in \mathcal{R}^{C_m \times H_m \times W_m}$ with $m$ denoting the order of the scale levels.

**Image Model.** Given a frontal view 2D image, a query-based transformer is applied as an image encoder to produce preliminary image BEV features by utilizing the IPM transformation. Then, these features are processed through the image backbone to generate the intermediate image BEV features. These hierarchical BEV features, $F_{bev,m}^I \in \mathcal{R}^{C_m \times H_m \times W_m}$, preserve the identical tensor shapes as LiDAR BEV features at the matching levels as shown in Figure 2.

### 3.2 Dual-Path Knowledge Transfer Mechanism

In learning-based models, whether CNNs or Transformers, hierarchical features at each layer contain different types of semantic information representation. It is well acknowledged that high-resolution shallow features in the early layers exhibit basic modality-specific perceptible properties [40], such as texture and edge for image pixels, and reflectivity for LiDAR points. Hence, a significant modality gap exists across these shallow modality features. As these features are progressively encoded into high-level abstract representations, the modality gap diminishes. Deep features at the bottom of a model have less about particular physical properties captured by different sensors and more about modality-agnostic contextual understanding relevant to the task. Based on this observation, we design a dual-path knowledge transfer mechanism for shallow and deep level features, aiming to provide precise LiDAR teacher features as a reference for image feature learning.

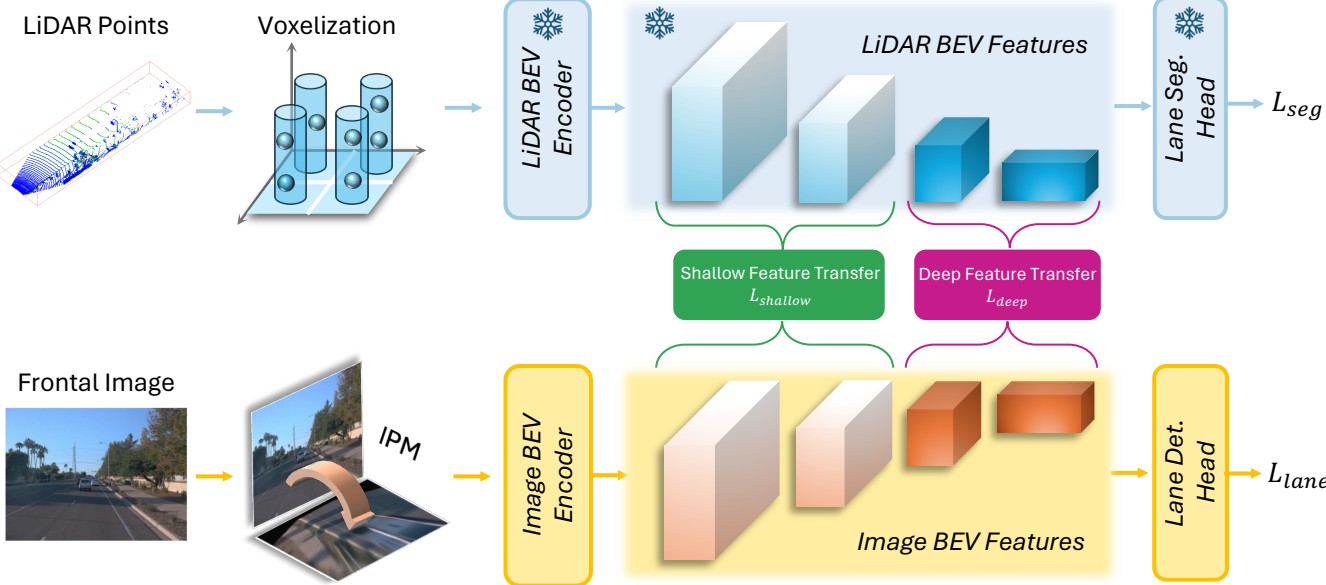

Figure 2: The schematic overview of our proposed 3D lane cross-modal knowledge transfer framework, LaneCMKT. We aim to bridge the learning discrepancy in BEV feature representation between the *LiDAR* and *Image* modalities for 3D lane detection. This approach enables the image model to learn spatial-awareness features guided by the LiDAR model. We propose a dual-path knowledge transfer mechanism to optimize the internal-model transfer learning for shallow and deep features. Noteworthy, the pre-trained LiDAR model is removed after training, our approach allows the image model to be enhanced without introducing additional computational overhead during inference.

**Shallow Feature Transfer.** Due to the inherent sparsity in LiDAR point clouds, the majority of LiDAR feature maps remain empty at the shallow level. Lanes with their slender and thin shapes only occupy a significantly small fraction of the overall scene. Therefore, these reasons result in the data imbalance of LiDAR teacher features. To allow image feature learning to focus on vital LiDAR geometric features of lane instances, rather than irrelevant background noises, we design an adaptive cross-modal scaling strategy for shallow feature transfer, including (1) *Instance Viability Scaling*; (2) *Length Adaptive Scaling*; (3) *Curvature Adaptive Scaling*.

*(1) Instance Visibility Scaling.* Although a monocular camera can perceive objects at farther distances than a LiDAR sensor, its field of view is susceptible to occlusion. Many incomplete lanes are captured by the camera, leading to poor scene condition understanding. LiDAR sensors, mounted at the top of a collection vehicle, provide a broader perceptive field and gather comprehensive road surface information. To leverage this advantage, we introduce an instance visibility scaling mask $G_{x,y}$ to convey 3D spatial information of visible lane instances from LiDAR to monocular image:

$$G_{x,y} = \begin{cases} 1, & \text{if } (x,y) \in R_c \\ \mu, & \text{if } (x,y) \in (R_l - R_c) \\ 0, & \text{if } (x,y) \notin R_l \end{cases} \quad (1)$$

where $x, y$ are 2D coordinates on BEV feature maps and $R_c$ and $R_l$ are the visible lane instance regions in the image and LiDAR shallow features, respectively. The visibility mask assigns different scaling weights for foreground regions across different modalities, which enables the image model to be aware of the modality perspective differences. While some portions of visible lanes may be recognized as False Positive (FP) from the front-view monocular perspective, our method encourages the image model to embrace such out-of-view spatial cues for holistic 3D scene learning.

*(2) Length Adaptive Scaling.* Lane lengths vary with traffic road conditions, yet the monocular image learning is insensitive to lane length variances. For example, shorter lanes, which occupy fewer image pixels in the frontal view, are described by fewer image features compared to longer lanes that span more pixels over an image. The image model processes all lane features equally. To address this problem, our cross-modal method aims to strengthen the image student model learning with sensitivity to variations in lane length. Specifically, shorter lanes require more guidance from LiDAR-based geometric features to counteract their limited image representation. We quantitatively represent the lane length as the number of lane instance pixels on the BEV map, an adaptive scaling mask $S \in \mathcal{R}^{H_i \times W_i}$ is introduced to adjust the image model's focus according to lane length variations:

$$S_{x,y}^k = 1 - \frac{n_k}{\sum_{k=0}^{K} n}, \ (x,y) \in O_k \quad (2)$$

where $n_k$ is the number of pixels of $k$-th lane instance in the foreground region $O_k$.

*(3) Curvature Adaptive Scaling.* Besides lane length invariance, another geometric property of lanes hindering the image model learning is the curvature. The monocular model can easily detect

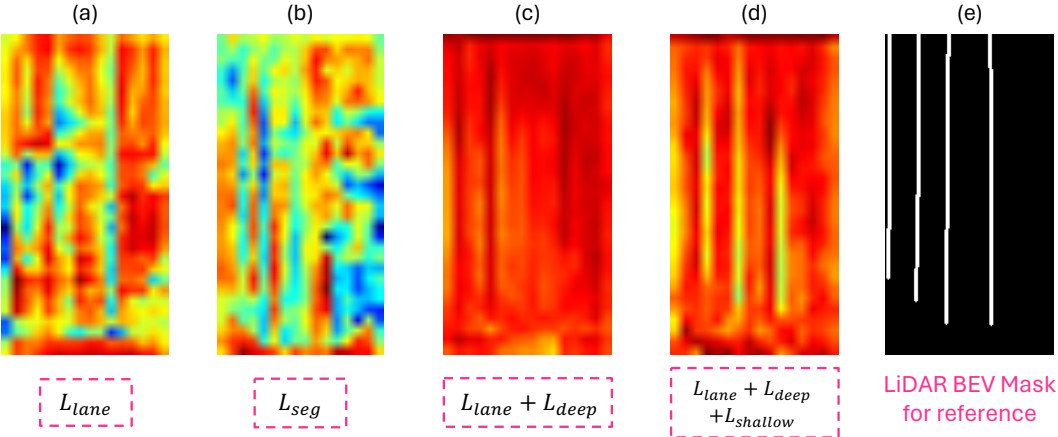

**Figure 3: Illustration of the pre-head BEV feature maps at the deepest level: from the image baseline model (a), from the LiDAR model (b), from the image model trained with deep feature transfer (c), from the image model trained with the dual-path knowledge transfer mechanism (d).** *The image BEV feature map is gradually becoming more distinctive for lane lines with our cross-modal knowledge transfer scheme.*

straight lane lines in highway driving scenarios. However, in urban scenes with many curved roads, due to the lack of accurate spatial cues, the image model produces ambiguous BEV features so as to degrade the detection performance. To enhance the image model's awareness of the lane curvature in knowledge transfer, we introduce a curvature scaling mask $U_{i,j}$ to emphasize the relevant geometric feature guidance from the LiDAR teacher model. Specifically, a lane curve can be parameterized as a quadratic polynomial as $y = a \cdot x^2 + b \cdot x + c$. To calculate the curvature of a lane, we quantize the absolute curvature at each discrete point along a lane and then average these values [3]:

$$\zeta_k = \frac{1}{I_k} \sum_{i_k=1}^{I_k} \frac{|2a_{i_k}|}{1 + (2a_{i_k} \cdot x_{i_k} + b_{i_k})^3} \quad (3)$$

where $x_{i_k}$ is the $x$ coordinate of a sample point and $I_k$ is the total number of sample points along a lane. The curvature adaptive scaling mask is formulated as:

$$U_{x,y}^k = \begin{cases} \sqrt{\zeta_k}, & \text{if } r_k < \varphi \\ 1, & \text{else} \end{cases}, \ (x,y) \in O_k \quad (4)$$

where $r_k$ is the scaled sum of squared residuals derived from the least squares fit of the quadratic polynomial of $k$-th lane and $\varphi$ is a quality control threshold.

In the end, combining all of the adaptive scaling masks, the shallow feature transfer loss can be written as:

$$L_{shallow} = \frac{1}{\alpha} \sum_m^M \sum_{k=1}^K \sum_{x=1}^H \sum_{y=1}^W G_{x,y}^k S_{x,y}^k U_{x,y}^k ||F_{bev,m}^I - F_{bev,m}^L||_2 \quad (5)$$

where $\alpha$ is the sum of the scaling mask weights.

**Deep Feature Transfer.** Deep features, as visualized in Figure 3 (a) and (b), contain fewer modality-specific intrinsic details but are rich in valuable global contextual embeddings. By encoding the modality features into a high-dimensional abstract representation,

this process eliminates modal variances and preserves the consensus semantic information relevant to a driving scene. To enable the image features to effectively mimic the modality-agnostic LiDAR geometric features, we employ the Mean Square Error (MSE) loss to facilitate the deep feature transfer:

$$L_{deep} = \sum_{m'}^{M'} ||F_{bev,m'}^I - F_{bev,m'}^L||_2 \quad (6)$$

where $m'$ is the index of the deep layers to perform feature transfer in LaneCMKT. Deep LiDAR BEV features are well-trained for BEV lane map segmentation which delineates lane layout within the global context. This spatial LiDAR pattern is vital to regularize ambiguous image BEV features. By conducting experiments, we observe that deep feature transfer contributes the most to a successful cross-modal knowledge transfer. Please head to our ablation study in Section 4.5 for more details.

### 3.3 Learning Objectives

**Training Loss Function for the Image Model.** Besides the knowledge transfer loss, we train the image student model with the original lane detection loss, which is calculated for each lane anchor as follows:

$$L_{lane} = L_{cls}(\hat{c}, c) + L_{reg}([\hat{x}, \hat{z}], [x, z]) + L_{vis}(\hat{v}, v) \quad (7)$$

where $L_{cls}$ is the cross entropy loss for lane category $c$, $L_{reg}$ is the $l_1$ norm regression loss for the offsets of anchor points along $x$- and $z$-axis, and $L_{vis}$ is the binary cross entropy loss for anchor point visibility $v$. We omit the index of lane anchors and other 2D auxiliary losses for brevity. We follow [38] to train the LiDAR teacher model.

**Training Loss Function for Knowledge Transfer.** In summary, the overall objective function is the sum of the shallow feature transfer loss (5), the deep feature transfer loss (6), and the 3D lane

**Table 1: Comparison with state-of-the-art methods on *OpenLane* validation set. The best values are marked in bold, and the second best values are marked in underline.**

| Method | F1(%)↑ | Cate Acc(%)↑ | X Err/C (m) ↓ | X Err/F (m) ↓ | Z Err/C (m) ↓ | Z Err/F (m) ↓ |
|---|---|---|---|---|---|---|
| 3D-LaneNet [8] | 44.1 | - | 0.479 | 0.572 | 0.367 | 0.443 |
| GenLaneNet [9] | 32.3 | - | 0.591 | 0.684 | 0.411 | 0.521 |
| Cond-IPM | 36.6 | - | 0.563 | 1.080 | 0.421 | 0.892 |
| PersFormer [5] | 50.5 | **92.3** | 0.485 | 0.553 | 0.364 | 0.431 |
| CurveFormer [1] | 50.5 | - | 0.340 | 0.772 | 0.207 | 0.651 |
| Anchor3DLane [12] | 53.1 | 90.0 | **0.300** | 0.311 | 0.103 | 0.139 |
| CurveFormer++ [2] | 52.7 | 88.1 | 0.337 | 0.801 | 0.198 | 0.676 |
| LaneCMKT (Ours) | **55.8** | 89.2 | 0.310 | **0.303** | **0.083** | **0.123** |

**Table 2: Comparison with state-of-the-art methods on *OpenLane* validation set under different challenging scenarios.**

| Method | All | Up & Down | Curve | Extreme Weather | Night | Intersection | Merge & Split |
|---|---|---|---|---|---|---|---|
| 3D-LaneNet [8] | 44.1 | 40.8 | 46.5 | 47.5 | 41.5 | 32.1 | 41.7 |
| GenLaneNet [9] | 32.3 | 25.4 | 33.5 | 28.1 | 18.7 | 21.4 | 31.0 |
| PersFormer [5] | 50.5 | 42.4 | 55.6 | 48.6 | 46.6 | 40.0 | 50.7 |
| CurveFormer [1] | 50.5 | 45.2 | 56.6 | 49.7 | 49.1 | 42.9 | 45.4 |
| Anchor3DLane [12] | 53.1 | 45.5 | 56.2 | 51.9 | 47.2 | **44.2** | 50.5 |
| SPG [43] | 53.7 | 46.2 | **59.2** | **54.8** | **49.8** | 41.9 | **52.1** |
| LaneCMKT (Ours) | **55.8** | **47.3** | 58.6 | 53.2 | 48.0 | 42.2 | 51.7 |

task loss (7). The training loss $L_{total}$ can be written as:

$$L_{kt} = \lambda_1 \cdot L_{shallow} + \lambda_2 \cdot L_{deep}$$
$$L_{total} = L_{kt} + \lambda_3 \cdot L_{lane} \qquad (8)$$

## 4 EXPERIMENTS

### 4.1 Dataset and Metrics

Our experiments are conducted using the *OpenLane* dataset [5], built upon the Waymo Open Dataset (WOD) [31]. This dataset is a comprehensive large-scale collection that includes 200k frames and 880k 3D lane annotations across six distinct driving scenarios and 14 lane categories. The point cloud data in WOD are captured by a top 64-beam LiDAR sensor at 10Hz frequency. We follow the evaluation metrics proposed in GenLaneNet [9], using a distance-based matching mechanism to evaluate the lane detection accuracy. A positive matching of each predicted lane with ground truth is counted when 75% of its covered y-axis reference points have a point-wise Euclidean distance under a pre-defined distance thresholding of 1.5 meters. For each matched predicted lane, we compute the quantitative metrics including F1-score, accuracy, and X/Z distance errors. The distance errors along each axis are calculated at the close (**C**) and far (**F**) ranges, respectively.

### 4.2 Implementation Setting

To prepare point cloud input, we first project LiDAR points onto the monocular image using a camera parametric transformation and then exclude points outside the frontal receptive field. We follow the same top-view region of interest by previous monocular 3D lane detection models with the xy range of $[(-10, 10), (3, 103)]$. We use PointPillar [17] as the LiDAR Encoder and Persformer [5] as the monocular encoder. To implement knowledge transfer across

two modalities, we set the matched cross-modal BEV feature maps of shape size of [256, 128], [128, 64] as shallow level features and [64, 32], [32, 16] as deep level features. The visibility scaling weight $\mu$ in Equation 1 is set to 10, and the curve fitting quality control $\varphi$ in Equation 4 is set to 0.2.

**Training.** For training the LiDAR teacher model, we use an Adam optimizer [15] with a weight decay of $1e^{-7}$. The learning rate is set to $2e^{-3}$ with the training epochs of 35 and the batch size of 6. When training the image student model in LaneCMKT, the teacher model is frozen. We train the image model with an Adam optimizer with a weight decay of $1e^{-3}$. The learning rate is set to $2e^{-3}$ with the training epochs of 38 and the batch size of 4. We apply the cosine annealing scheduler [23] to periodically adjust the learning rate with $T_{max}$ of 8. In loss functions of Equation 8, $\lambda_1, \lambda_2, \lambda_3$ are set to 1, 64, 1, respectively.

### 4.3 Quantitative Results

**Main Results.** We present our experimental results on the validation set of *Openlane*. To make a comprehensive comparison in this study, we categorize two mainstream monocular 3D lane detection approaches of "decoded from IPM BEV features" and "decoded from back-projected spare image features" to the BEV-driven methods. As shown in Table 1, our LaneCMKT model significantly outperforms all prior approaches by notable margins. It achieves a remarkable improvement over the image baseline, Performer [5], by a 5.3% increase in F1 score. We observe reductions in the distance errors by 0.175m/0.250m and 0.281m/0.308m in the X and Z axes for the close/far range. Furthermore, our method surpasses the BEV-driven SoTA method, Anchor3DLane [12], by 2.7% in F1 score and shows improvements in distance errors by 0.008m in the X axis

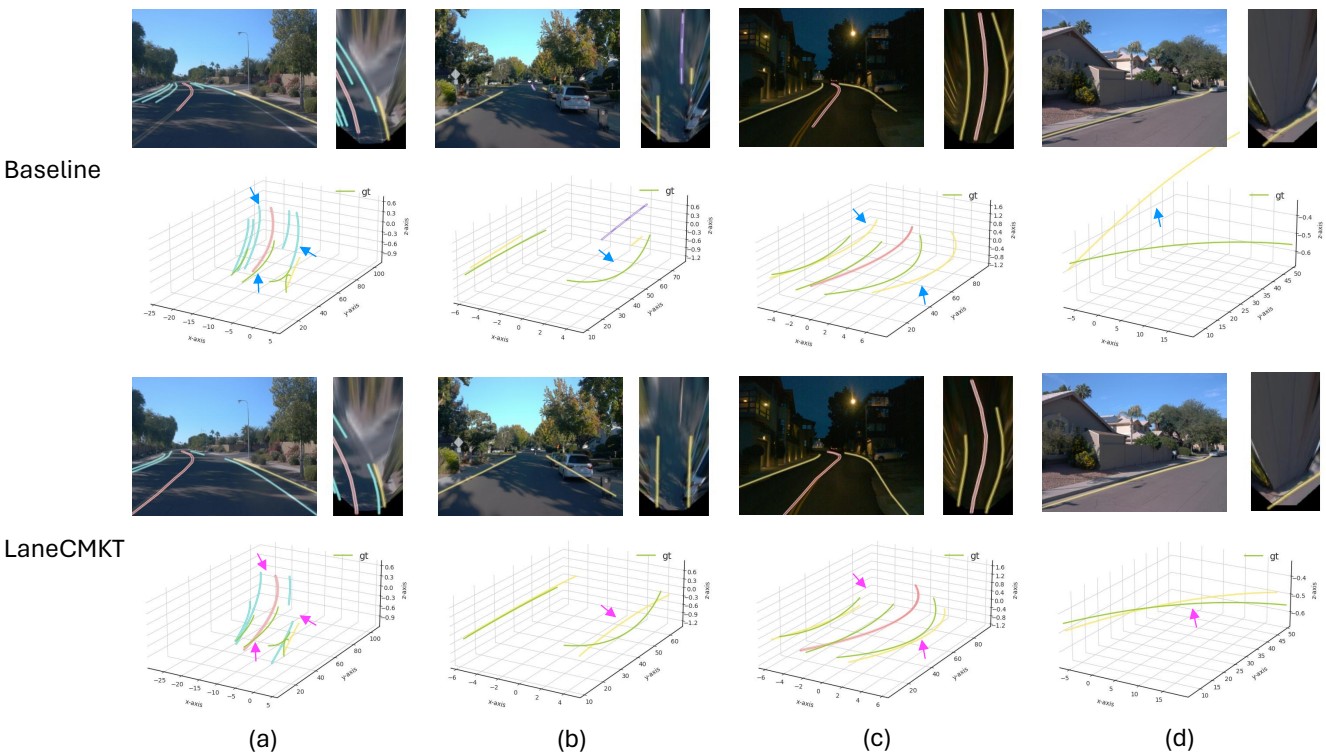

Baseline

LaneCMKT

(a)                                    (b)                                    (c)                                    (d)

**Figure 4: Qualitative analysis on *OpenLane* validation set. In each case, 3D lane prediction is projected to the frontal view and IPM BEV perspectives for clear comparisons, and the prediction and ground truth are visualized in 3D space. Differences in predictions between our LaneCMKT and the baseline are highlighted using colored arrows. Please zoom in to view the details.**

of the far range and 0.020m/0.016m in the Z-axis of the near/far range, respectively.

**Evaluation Under Difference Driving Scenarios.** Apart from the main experimental evaluation on *OpenLane*, we also conduct experiments across various challenging scenarios with F1 scores reported accordingly. We add SPG [43] into Table 2 to make a fair comparison, because it has a SoTA performance across these challenging driving scenarios to our best knowledge. Its robustness comes from the strategy of re-utilizing image features to refine coarse lane anchors during additional post-processing stages. However, this multi-stage method complicates the detection workflow and reduces efficacy during model inference. Our model surpasses SPG by 1.1% in the common uphill/downhill scenarios and achieves comparable performances for other scenarios without introducing extra processing steps or learnable model parameters. This suggests that our method enables the monocular image model to elegantly and effectively learn vital spatial awareness to confront challenging driving scenarios.

## 4.4 Qualitative Analysis

We present a qualitative comparison between LaneCMKT and the image baseline in Figure 4. We validate the performance of our method by challenging scenarios, including uphill elevation *(a)*, occlusion *(b)*, low-illumination condition *(c)*, and irregular horizontal

lanes in the frontal view *(d)*. The visualization of predictions in 3D space illustrates that our method achieves more robust and accurate 3D lane predictions in these difficult perception environments compared to the image baseline. This indicates that the image baseline benefits from the spatial awareness and improved generalizability learned from the LiDAR modality in our proposed cross-modal knowledge transfer training scheme.

## 4.5 Ablation Study

We follow the previous works [5, 12, 24, 26] to conduct the ablation study on *OpenLane-300*, a subset of *OpenLane*. Other experimental configurations remain the same as in Section 4.2.

**Effect of Dual-Path Knowledge Transfer Mechanism.** We investigate the effect of image feature learning conducted at the shallow and deep layers within our designed dual-path knowledge transfer mechanism. As illustrated in Table 3, the deep feature transfer contributes the most to the accuracy of lane detection and localization. The deep features are favorable to encode lane intrinsic spatial insights without containing much modality-specific information, so monocular image features can directly benefit from the geometric contextual embedding from the LiDAR teacher model. Although using only shallow feature transfer does not yield notable performance gains, incorporating shallow LiDAR supervision makes lane instance features more discriminative on the pre-head

**Table 3: Effect of the dual-path knowledge transfer mechanism at distance matching threshold of 1.5 meters.**

| Method | Shallow | Deep | F1(%)↑ | X Error (C/F) ↓ | Z Error (C/F) ↓ |
|---|---|---|---|---|---|
| Baseline | ✗ | ✗ | 58.12 | 0.403/0.381 | 0.121/0.152 |
| LaneCMKT | ✓ | ✗ | 58.87 | 0.401/**0.365** | 0.121/0.147 |
| | ✗ | ✓ | 59.02 | 0.399/0.378 | 0.118/0.146 |
| | ✓ | ✓ | **59.85** | **0.362**/0.373 | **0.116/0.139** |

**Table 4: Effect of the dual-path knowledge transfer mechanism at distance matching threshold of 0.5 meters.**

| Method | Shallow | Deep | F1(%)↑ | X Error (C/F) ↓ | Z Error (C/F) ↓ |
|---|---|---|---|---|---|
| Baseline | ✗ | ✗ | 37.71 | 0.306/0.300 | 0.123/0.150 |
| LaneCMKT | ✓ | ✗ | 38.23 | 0.309/0.295 | 0.117/0.137 |
| | ✗ | ✓ | 40.87 | 0.301/0.285 | 0.114/0.139 |
| | ✓ | ✓ | **41.61** | **0.297/0.275** | **0.104/0.125** |

BEV maps as demonstrated in Figure 3. Therefore, the shallow and deep level feature transfers collectively maximize lane detection performance. Additionally, we evaluate our method performance with more stringent distance matching criteria as shown in Table 4. The performance improvement is augmented from 1.73 to 3.45, which underscores that the robustness and effectiveness of our method persist even under high spatial requirements. This enhancement also demonstrates that our method enhances the monocular image model with superior 3D spatial awareness.

**Effect of Adaptive Scaling Strategies.** We systematically validate the effect of the adaptive scaling strategies designed to bridge the modality gap by encouraging image BEV feature learning to focus more on critical 3D lane geometric cues. Table 3 shows that combining shallow and deep feature transfers yields a significant performance enhancement compared to employing shallow feature transfer alone. Therefore, we start with deep feature transfer and progressively incorporate the instance visibility scaling, the length adaptive scaling, and the curvature adaptive scaling. Table 5 demonstrates that each component makes a gradual improvement for lane detection, and they collectively achieve optimal performance. Among the scaling strategies, lane curvature scaling contributes most significantly to the improvement, which demonstrates that this geometric property scaling enables the image feature to pay more attention to the complex lane shape under the spatial supervision of the LiDAR model.

**Effect of Different Image Encoders.** We study the impact of various image encoder scales (EfficientNet-B3, B5, B7) on the detection performance of our LaneCMKT, investigating the role of the image encoder in the context of cross-modal knowledge transfer. As presented in Table 6, incorporating our method improves the detection performance across various image baselines with image encoders of different scales. Remarkably, a smaller-scale monocular model (EfficientNet-B5) outperforms a larger-scale model (EfficientNet-B7) when applying our LaneCMKT training scheme. This validates the advantage of leveraging geometric features extracted from the LiDAR teacher model enables the image student model to beneficially

**Table 5: Comparisons of the adaptive scaling strategies. *IVS* denotes Instance Visibility Scaling, *LAS* denotes Length Adaptive Scaling, and *CAS* denotes Curvature Adaptive Scaling.**

| *IVS* | *LAS* | *CAS* | F1(%)↑ | X Error (C/F) ↓ | Z Error (C/F) ↓ |
|---|---|---|---|---|---|
| ✗ | ✗ | ✗ | 59.02 | 0.399/0.378 | 0.118/0.146 |
| ✓ | ✗ | ✗ | 59.07 | 0.385/0.377 | 0.118/0.140 |
| ✓ | ✓ | ✗ | 59.13 | 0.371/0.378 | 0.119/**0.139** |
| ✓ | ✓ | ✓ | **59.85** | **0.362/0.373** | **0.116/0.139** |

**Table 6: Comparisons of our method with different image encoders.**

| Method | F1(%)↑ | X Error (C/F)↓ | Z Error (C/F)↓ |
|---|---|---|---|
| EfficientNet-B3 | 58.62 | 0.390/**0.361** | 0.116/**0.145** |
| EfficientNet-B3 + LaneCMKT | **59.08** | **0.368**/0.371 | **0.115**/0.148 |
| EfficientNet-B5 | 57.46 | 0.421/0.401 | 0.123/0.158 |
| EfficientNet-B5 + LaneCMKT | **58.63** | **0.378/0.374** | **0.120/0.152** |
| EfficientNet-B7 | 58.12 | 0.403/0.381 | 0.121/0.152 |
| EfficientNet-B7 + LaneCMKT | **59.85** | **0.362/0.373** | **0.116/0.139** |

learn 3D-aware features with reduced model complexity. It is observed that EfficientNet-B3 surpasses other larger monocular image models which are prone to overfitting to the large-scale lane dataset. Nonetheless, upon introducing our method, the performance of the model using the EfficientNet-B3 encoder is less accurate compared to that using the EfficientNet-B7 encoder. This suggests that the performance gain of cross-modal knowledge transfer depends on the interpretation ability of the student encoder. A more powerful image encoder, such as EfficientNet-B7, allows the student model to adapt more effectively to the feature distribution variations based on cross-modal knowledge transfer during the training phase.

## 5 CONCLUSION

In this study, we present a compact cross-modal knowledge transfer training framework, LaneCMKT which utilizes spatial insights from a LiDAR-based teacher model to guide image representation learning of the monocular student model for 3D lane detection. Our approach includes a dual-path feature transfer mechanism for shallow and deep level features to adaptively distill the geometric cues from the teacher model. Extensive experiments substantiate the effectiveness of our 3D-to-2D knowledge transfer framework with the remarkable improvement in detection performance compared to the baseline and other BEV-driven SoTA methods.

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
