# OpenReview forum: "LaneCMKT: Boosting Monocular 3D Lane Detection with Cross-Modal Knowledge Transfer"
_acmmm.org/ACMMM/2024/Conference — MM2024 Poster_

### Official Review · Reviewer_S7ik · 2024-05-25

**Rating:** 6
**Confidence:** 4

**Summary:**

Very good! Lane line detection is inherently a task that requires lightweight real-time performance. Instead of using conventional direct detection of multi-source data, this work uses migration learning to significantly improve the detection at little additional cost.

**Strengths:**

1. By using a transfer learning approach rather than learning directly from multiple data sources, the network can be effectively upgraded at minimal additional cost.

**Limitations:**

Ablation study of IVS and LAS showed no significant improvement

**Suitability:**

3

---

### Official Review · Reviewer_KvBr · 2024-05-26

**Rating:** 4
**Confidence:** 3

**Summary:**

This paper presents a compact cross-modal knowledge transfer training framework, LaneCMKT which utilizes spatial insights from a LiDAR-based teacher model to guide image representation learning of the monocular student model for 3D lane detection. The proposed approach includes a dual-path feature transfer mechanism for shallow and deep level features to adaptively distill the geometric cues from the teacher model.

**Strengths:**

In this paper, a new training method for monocular 3D lane line detection is proposed, which uses a pre-trained 3D laser point cloud model to train a lane line detection model based on monocular images. At the same time, the authors deeply analyze the problems existing in cross-modal knowledge transfer and propose a dual-path transfer algorithm. This method solves the shortcomings of the traditional IPM method, and also makes full use of multi-modal information, and the model performance is improved compared with the benchmark method.

**Limitations:**

1. How are the weight of each loss selected? An analysis should be provided.
2. In the visual comparison, only a comparison with the baseline is provided, visual comparisons with state-of-the-art (SOTA) methods should be added.
3. The capitalization in the references should be consistent, such as the ECCV conference in references 4 and 5.

**Suitability:**

2

---

### Official Review · Reviewer_Qgsm · 2024-06-01

**Rating:** 3
**Confidence:** 3

**Summary:**

This paper proposes a new monocular 3D lane detection model, called LaneCMKT, by distilling cross-modal knowledge from a pretrained LiDAR 3D lane detection model. In detail,  the authors design a dual-path knowledge transfer mechanism to accurately extract multi-layer geometric knowledge from the LiDAR teacher model. Besides, an adaptive scaling strategy is proposed to mitigate the modality discrepancy at shallow level features. The proposed model achieves SOTA performance on a large-scale lane dataset, OpenLane.

**Strengths:**

1. The paper is well-written and easy to understand.
2. The proposed method achieves competitive detection performance.
3. Ablation studies prove the effectiveness of the proposed designs.

**Limitations:**

1. Lack of novelty. Distilling LiDAR knowledge to camera-only models has been well studied. The method presented in this paper is only a small offset from the existing work.

2. The authors are encouraged to try different LiDAR teacher models and camera student models to prove the versatility of the proposed method.

**Suitability:**

3

---

### Meta-Review · Area_Chair_d7Mm · 2024-06-30

**Recommendation:** Accept (Poster)
**Confidence:** 5

**Metareview:**

This paper proposes a tailored distillation method to enhance monocular 3D lane method with LiDAR feature. To address the misalignment in camera and LiDAR BEV feature, a dual-path knowledge transfer module is proposed to alleviate it and better transfer the 3D geometric clues from LiDAR feature. Experiments on OpenLane show its effectiveness over current BEV-driven SoTA methods.

All three reviewers agree on the acceptance of this paper for its effectiveness validated on the widely used benchmark. Some concerns such as lack of novelty and experiment details were addressed well in the rebuttal. After considering all these effort, AC agrees with reviewers that the paper should be accepted. Please revise the paper accordingly based on comments and incorporate new experiments provided in the rebuttal.